# Decentralized Processing Performance of Fruit and Vegetable Waste Discarded from Retail, Using an Automated Thermophilic Composting Technology

**Florin Nenciu** [1,*], **Iustina Stanciulescu** [2], **Horia Vlad** [2], **Andrei Gabur** [2], **Ovidiu Leonard Turcu** [3,*], **Tiberiu Apostol** [4], **Valentin Nicolae Vladut** [1], **Diana Mariana Cocarta** [4] and **Constantin Stan** [4]

1   National Institute of Research—Development for Machines and Installations Designed for Agriculture and Food Industry—INMA Bucharest, 013811 Bucharest, Romania; vladut@inma.ro
2   Electronic Waste Management SRL, 137370 Potlogi, Romania; iustina@weee.ro (I.S.); horia@weee.ro (H.V.); andrei@weee.ro (A.G.)
3   Faculty of Economic Sciences, Vasile Alecsandri University of Bacau, 600115 Bacău, Romania
4   Faculty of Energy Engineering, University POLITEHNICA of Bucharest, 060042 Bucharest, Romania; tiberiuapostol80@gmail.com (T.A.); diana.cocarta@upb.ro (D.M.C.); stan.constantin@yahoo.com (C.S.)
*   Correspondence: florin.nenciu@inma.ro (F.N.); ovidiuturcu@ymail.com (O.L.T.)

**Abstract:** Food waste generation is increasing at an exponential rate, affecting the environment, food security, and causing major economic issues worldwide. The main aim of the current research is to investigate a novel composting technology that is still in its early stages of development. The proposed composting technology combining thermophilic composting with the use of advanced automated processing reactors. Starting from a qualitative and quantitative analysis of the waste generated at retail-stores, the most significant difficulties associated to waste management as well as the main characteristics of the discarded waste were identified. The findings allowed to design and evaluate the real operating performance of an automated thermophilic composting prototype (working in a decentralized regime), with the goal of delivering a faster processing system, improving operational efficiency, reducing expenses, and lowering environmental impacts. The proposed operating technique showed a high capacity for pathogens and seeds removal, the waste input mass reduction of 88%, and efficiency in food processing (2235 kg of fruits and vegetables in a 14-days timeframe).

**Keywords:** food waste; composting system; organic waste management; decentralized composting

## 1. Introduction

Food waste has been one of the main challenges over the last decade, since discarding large amounts of organic waste shown to generate severe impact on the environment and human health. Pathogenic microorganisms and toxic leachates reach agricultural soil and water sources, producing greenhouse gases and causing a number of long-term health negative effects. Composting is an ecological management strategy which is used for decades for processing organic waste in accordance with global sustainable development goals, at various levels of efficiency and technological advancement [1]. One-third of the food produced worldwide for human consumption is being wasted every year [2,3]. Despite the fact that many municipal waste management operators still regard organic waste as a major challenge, these materials are in fact important renewable resources, that may be converted into valuable products (such as compost) using different novel processing technologies. Compost is used for a variety of purposes, including mushroom cultivation, mulching, improving soil structure in orchards, and producing brew compost tea [4]. A

reduction of worldwide food losses and the use of vegetable waste to produce valuable agricultural fertilizers might be a solution to increase future food security [5,6].

### 1.1. Issues of Traditional Waste Management Systems for Fruit and Vegetable Waste Discarded from Retail Outlets

Fruits and vegetables account for up to 50% of all food waste discarded from grocery stores, supermarkets, and food courts. The difficulties encountered for efficient processing of these food wastes are caused by certain distinct characteristics that have always limited large-scale application. Common difficulties usually include high moisture content, reduced porosity that makes air penetration difficult, rapid oxidation/putrefaction, faster natural microbial activity, the risk of developing pathogens, unpleasant odors, or attracting insects. Fruits and vegetables incur high levels of loss given their high perishability, their limited shelf life, improper handling, or maintenance at unsuitable temperatures. Furthermore, their moisture level often exceeds 80% vol. [7]. Many studies have found several difficulties in composting fruit and vegetable waste due to the high moisture content and low porosity during processing, which can lead to anaerobic decomposition conditions, having negative consequences in terms of quality loss, odor generation, and pathogens [8,9].

The centralized processing is the most used municipal organic waste management system. In this approach, a regional operator collects and processes fresh waste on a large specific area using simple and expensive techniques and producing odors, leachate, and anaerobic degradation of organic matter [10,11]. Centralized municipal waste management systems may lead to several operating difficulties and challenging situations. Some examples in this regard are high processing and transportation costs, need to continuously upgrade the treatment methodologies in accordance with the composition of the waste, high risk of generating pathogens dangerous for the environment and human health, as well as difficult integration of the products as fertilizers in the agricultural circuit [12]. On the other hand, decentralized solid waste management is a system meant to ensure a sustainable and safe environment by processing smaller quantities of waste at the source.

Composting is an aerobic, microorganism-mediated fermentation process, performed particularly for the solid fraction, used to transform organic materials into more stable compounds. The product obtained following processing is the compost, a highly valuable product for agriculture use, since it can improve a soil's physical, chemical, and microbiological qualities [13,14]. A more efficient technique for processing organic waste in order to obtain qualitative compost is the thermophilic composting in enclosed systems, using specially designed bioreactors. Using novel bioreactor designs, equipped with automation means and sensors, which can operate autonomously, larger amounts of waste can be processed in a limited space, being able to manage any form of organic waste (meat, animal manure, biosolids, and food scraps) [15,16]. Under aerobic conditions, thermophilic composting transforms organic solid waste into agricultural resources such as organic fertilizer or soil amendment, while also achieving pathogen inactivation and waste volume reduction. When compared to other conventional techniques, thermophilic composting has the benefits of reduced prices, simplicity of operation, and less residues produced [17–19].

### 1.2. Main Findings and Concerns Related to Composting Processes

Research studies have suggested the need of supplemental aeration by introducing forced ventilation mechanisms to improve processes [20–23], given that in the classical processing regime, oxygen was found to be consumed within two hours after the mechanical turning operation [24,25]. The composting efficiency is determined by the microbial activity and it is affected by a variety of variables such as biomass properties, oxygen, temperature, moisture, and waste structure [26,27].

The composition of the microbial population changes continuously during the process, as a result of the changes that occur in the composting material parameters and environment. A temperature range of 20–55 °C is considered by many authors [28,29] to be the most favorable to the process efficiency, because a further increase will result in an

inactivation of the microorganisms and a drop-in of the process rate. Compost organisms can survive with 5% vol. oxygen. However, a reduction in oxygen levels below 10% vol. in the gas phase can lead to a lower process rate and odors, as anaerobic conditions are usually created [30].

Bacterial species diversity generally refers to the number of different species present in an environment and the relative evenness of the number of individuals found from each species. Species diversity is usually not correlated with biological productivity. However, the biological productivity is correlated with the stability of the community [31], being an essential attribute in composting processes. A study of continuously thermophilic composting [32] was conducted in an incubator, to determine the effect of temperature impact on the bacterial species diversity. The experiment concluded that a high species diversity is needed for proper bacterial population stability and metabolic versatility, while the maximum desirable temperature for continuously thermophilic composting is found to be maximum 60 °C. In order to reduce composting time and increase the quality of the compost, the ratio between leaves and straw should be as high as possible, at least 2:1 [33], alternatively the composting process will take longer. The poorly-matured compost has to be applied far in advance of cultivating a crop to allow for supplementary biological activities in the soil [34,35].

Pathogen presence is one of the main concerns whether they are considered human pathogens or pathogens that can affect the plant health. Human pathogens contained in the compost may cause illness when it is handled or when contaminated plants are consumed, while crops may be severed affected when they come in contact with an infested compost added as fertilizer. Pathogen removal is frequently ensured by extending the residence period of the waste in the process at a specific temperature level. Dry matter content is preferred to be around 60–70% considering the reduced handling and transportation expenses [36,37].

A study that assessed the effects produced by active and passive aeration on composting household biodegradable wastes, in a decentralized approach [13], found that active aeration is the most effective approach for residential organic waste treatment especially for reducing the unpleasant odors and obtaining financial savings by reducing processing time. Other research studies have approached the effect produced by the intermittent rotation with natural aeration for enclosed bioreactors [38–40], or the combined continuous aeration accompanied by rotary mixing process for reducing the active period [15] for mixtures of fruits, vegetables, and chicken manure. Results showed that the continuous aeration-rotation significantly reduced the active phase period to 4.5 days, increased the compost temperature (Tc) to 60 °C after three days of operation, and remained at 50–65 °C for approximately three consecutive days (thermophilic stage). The temperature range in thermophilic regime confirms the destruction of the pathogens in the compost, as well as the reduction of the processing time.

The mechanism for obtaining high-quality compost from fruit and vegetable wastes was also investigated [41] to establish the relationships created between microorganisms and physicochemical parameters. The mathematical model highlighted the fact that strongest positive relations are created between moisture and carbon contents, moisture and coliform bacteria, moisture and C/N ratios. Microbial communities are continually changing during the composing process, both in number and in species variety [42]. Regarding various indicators that can provide information about the process, previous research studies [43] observed that an increase in the number of microorganisms during composting indicates a more extensive and efficient biodegradation, and when the number of microorganisms decreases rapidly, it indicates that the compost is mature and stable.

*1.3. The Main Aim and Expectations in the Framework of the Current Research Study*

The current research proposes a novel approach for establishing more efficient control and management of the biological degradation process by developing a successful operational composting system for high moisture content waste.

The decentralized composting approach is designed to reduce total processing costs, as well as to minimize the space and time required to decompose food waste. Over time, this kind of approach will partially relieve authorities of a range of activities associated with organic waste management [44]. This research intends to optimize composting technology operational parameters in order to create a system that maximizes mass and volume reduction while also shortening the overall composting time, bringing financial and environmental benefits. In this regard, an experimental organic waste composting prototype has been considered, which has been customized for the processing of discarded fruits and vegetables from retail stores, and equipped with a series of sensors and process control means to get the clearest results.

## 2. Materials and Methods

*2.1. Research Approach and Legislative Considerations Outlining the Working Methods*

The difficulties experienced by the public authorities to effectively manage the expanding amounts of organic waste, as well as the increasing volumes of organic waste delivered to landfills, shifts to a large extent the responsibility on waste generators. Retail stores already encounter a progressive increase in disposal prices, imposed to discourage waste generation; therefore, several supermarket chains are already implementing pilot decentralized treatment systems. Even though all stakeholders aim to reduce waste quantities, supermarkets have a different perspective than authorities, because their major goal is to lower the costs of organic waste treatment using all available options, and not to obtain a superior quality of the compost. At the moment, there is no national policy in Romania to assist composting projects for retail sector, nor a public infrastructure that can be utilized to efficiently manage organic waste. In addition, there are no clear regulations so far governing the quality of compost being produced, the legislation in force is vague and does not yet have implementing norms. Most likely, no matter how high is the final compost quality, under current legal conditions the compost will reach the landfill anyway.

Therefore, when designing the methodology and the automatic decentralized composting system used in the processing of fruit vegetable waste, we consider especially the needs of the stores and less the compost quality. The retailers refused to use inoculum and enzymes, since the compost superior quality does not bring any benefits to the stores, and increase their acquisition and operating costs. The retail stores expectations regarding fruit and vegetable wastes were reducing the mass and volume of waste, reducing operating costs, reducing unpleasant odors, eliminating pathogens and water that could leach from the waste mass, and temporary waste inactivation/stabilization.

Any improvement process requires very good knowledge of the processed material; therefore, the first phase consisted in identifying the characteristics and quantities of organic waste, discarded from grocery shops and supermarkets. The evaluations regarding their quality allowed us to make projections about the way how the composting process should be carried out in order to optimize the operations, in accordance with the proposed objectives.

The second phase began with a modified composting equipment manufactured for decentralized organic waste composting, which was upgraded with certain customization and parameter monitoring tools. Considering that the process is not customized to obtain the best compost quality but on performance indices designed by retail stores, we called the process "pre-composting", although introducing small functional changes can increase the quality indicator. Three pre-composting process determinations have been conducted in the thermophilic regime, while the results were evaluated by examining the product obtained after 15 days of composting. Several important performance indicators

were used to evaluate the compost and the composting process, including mass, volume, and volatile solids reduction, C/N ratio, and temperature.

The main stages of the case study involving efficient management practices of fruit and vegetable waste disposed from retail stores is presented in Figure 1.

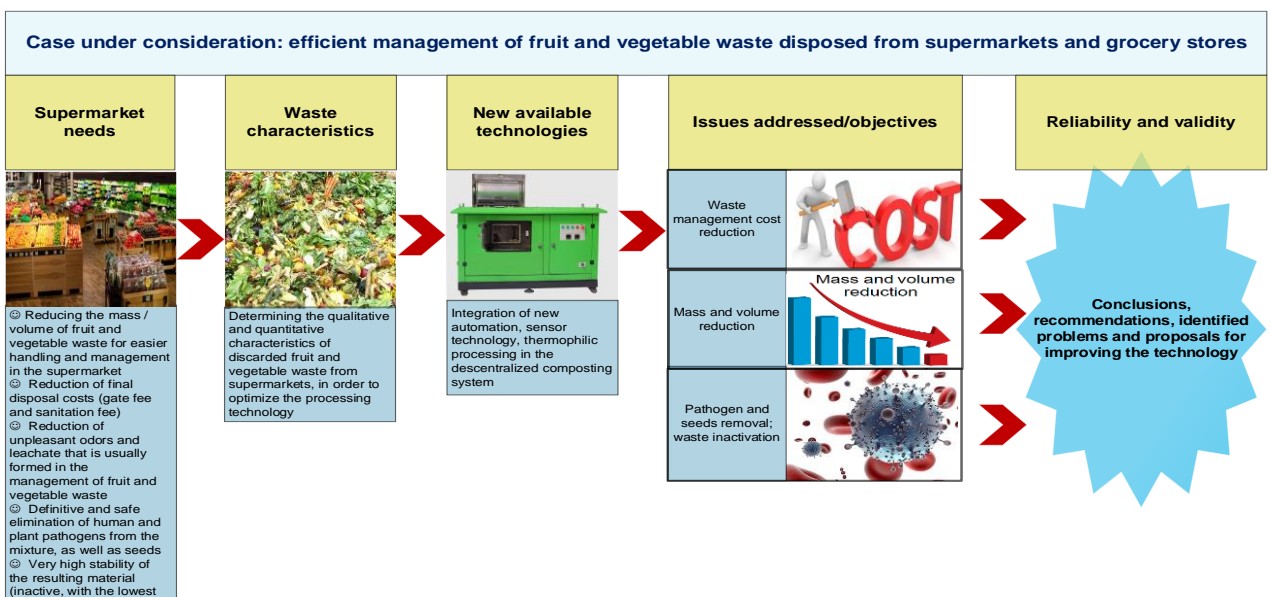

**Figure 1.** The main stages of the research study involving efficient management practices of fruit and vegetable waste disposed from retail stores.

## 2.2. Description of the Composting System and Experimental Design

The most important part of a composting technology is the processing reactor, which is the subassembly where organic wastes are transformed into compost. The reactor functioning in intensive thermophilic regime is not only a waste container, but also performs the aeration, mixing, heating, dehydration, inoculation, and the control of the process.

Figure 2 outlines the operating principle used for the automatic equipment used for decentralized composting of organic waste.

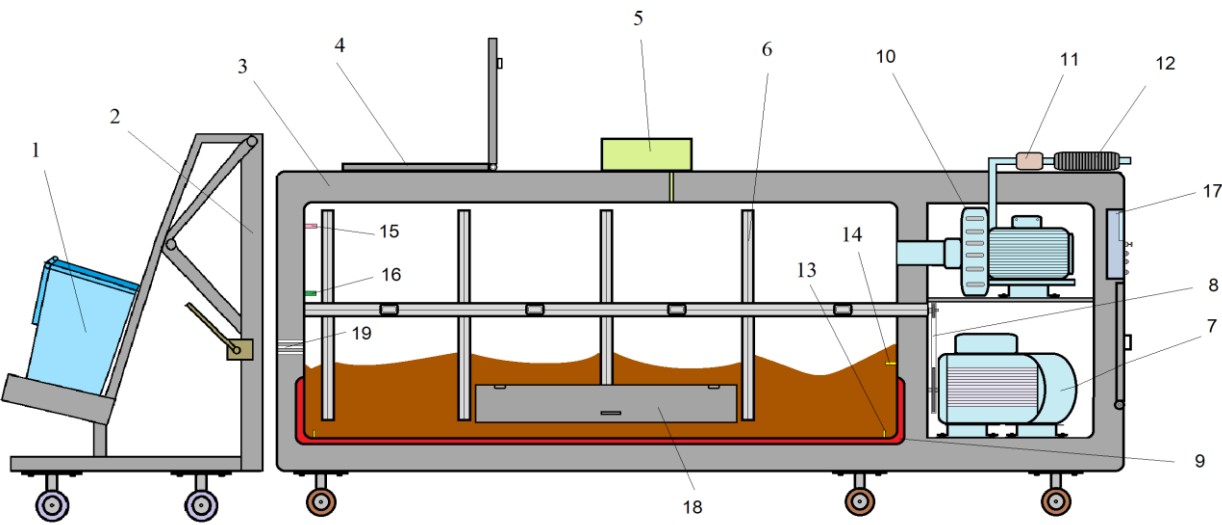

**Figure 2.** Operating principle of the automatic equipment used for decentralized composting of organic waste: 1—recycle bin, 2—hydraulic actuation tipping equipment, 3—composting reactor, 4—the feeding door, 5—inoculum/enzyme dosing, 6—blades mechanism, 7—electric motor, 8—driving mechanism, 9—ceramic heating system, 10—air pump, 11—gas analyzer, 12—deodorizing filter,

13—heater temperature sensor, 14—waste temperature sensor, 15—moisture sensor, 16—volume sensor, 17—command-and-control panel, 18—outlet door, 19—air inlet.

The recycle bins (1) containing sorted organic waste are emptied with the use of a hydraulic actuation tipping equipment (2) inside the composting reactor (3), via the feeding door (4). The composting system is provided with a dispenser that is used for inoculum/enzyme dosing (5), and a rotating blades mechanism applied for mixing and aerating the processed material (6), which is driven by an electric motor (7), connected to a driving mechanism (8). The temperature inside the equipment is rapidly increased with a ceramic heating system (9), which has the role of creating the needed conditions for thermophilic composting and to dehydratase the fruits and vegetables, allowing processing of a larger volume of organic material. The excess air moisture is being absorbed by an air pump (10), which is provided with gas analyzer (11) and a deodorizing filter (12). The sensors associated with the reactor are used to detect the temperature of the heater (13), detect the temperature of the material being processed (14), a sensor monitors air humidity (15), and the volume of material inside the composting equipment (16). The command-and-control panel (17) is used to customize the processing parameters depending on the input material, being able to accurately set temperature ranges, adjust the moisture level when the air blower starts functioning, and the volume ranges for starting and stopping operating the equipment. When the process is complete, the outlet door (18) is used to empty the automatic composter, allowing the processing cycle to resume. The composting reactor is never completely emptied, it is designed so that a quantity of 2–10% of processed compost to remain in the equipment, in order to contribute to a faster inoculation of the new batch of processed organic material.

Therefore, the experiments were performed in a closed industrial hall, without any heating or air conditioning systems. The equipment used for the experiments is a customized automated composter, which has been upgraded with a series of sensors and functions to better respond to the evaluation process of internal processes.

The mixing and aeration blades have a specific constructive shape, which also allows the crushing/shredding of the vegetal material in the process, as can be seen in Figure 3.

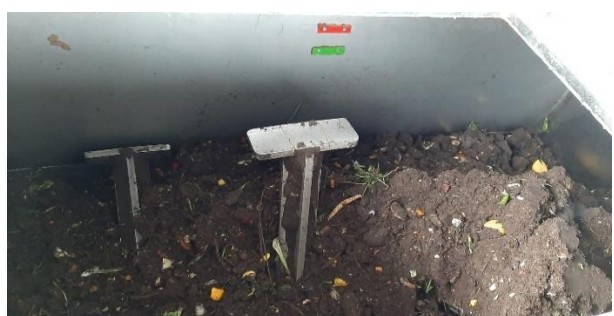 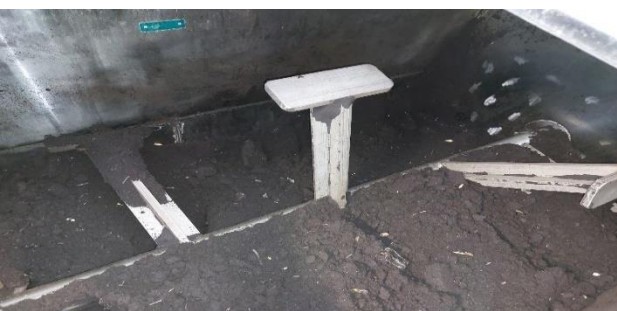

**Figure 3.** Blade design with a role in aeration, mixing, and shredding of processed fruits and vegetables.

The designed capacity of the composting reactor is 250 L. The reactor is constructed of specially treated steel for increasing resistance, considering the corrosive working conditions. The opening located at the upper part of the reactor (having the size of 1 m²) is used for supply with vegetable waste, using either a conveyor or a hydraulic system. The mixing action is carried out by the rotating blades mechanism made of stainless-steel bars, welded on a shaft. Its operation is driven by an electric motor, the speed and direction of rotation of the blades can be customized electronically. The ceramic electric heater has the role of speeding up the composting process, by maintaining the organic matter in thermophilic conditions. The most important role, however, is the rapid dehydration of fruits and vegetables, for the processing of large volumes of organic material. For this, the heater operates continuously until the material reaches the set temperatures. The sensors, installed at the bottom of the composting reactor, were used to monitor and control the

temperature, and the data were logged with a programmable logic controller (PLC) and a Graphic operational Controler (GOC43). Ambient temperature was measured with a Thermo-Hygrometer DMA033 type.

The vapors formed in the equipment are evacuated by the air extractor, while the dry air enters the reactor through 10 holes (4 mm each), located on the opposite wall. Moisture removal is conducted by the moisture sensors, triggering the air pump operation, in order to keep the compost moisture in the parameters. The sensors installed inside the gas analyzer, located on the exhaust pipe, have the role of maintaining the processing in aerobic conditions. If the sensors detect some gases such as methane or sulfur dioxide, the program changes its operating mode and return the process to aerobic conditions (heating and moisture extraction are intensified).

The measured parameters were recorded every 10 s, averaged for every 10 min period and saved in the data logger.

This paper does not evaluate the automation characteristics; however, some of the features are worth mentioning, given that they contribute to increasing the efficiency of composting and reducing operating costs. First, the four types of sensors (heaters temperature sensor, compost temperature sensor, compost moisture sensor, level sensor) start and finish all operations autonomously (heating, mixing, aeration, water evacuation), reducing employee intervention. It also improves the time required for treatment, given that the system instantly announces when the process is complete. The air outlet sensor provides safety because it announces when the gases specific to the anaerobic digestion process are produced and in consequence automatically intervenes to improve aeration.

### 2.3. The Waste Loading Mode in the Thermophilic Regime Approach

In the 16 days associated with pre-composting, the amount of fruit and vegetables introduced for processing is gradually increased, so that microorganisms can inoculate the mass of material at the highest efficiency. The moisture level up to which the material dehydrates has been set to 20%, because through a daily intake of fresh material the moisture variation interval is in optimal values this way. Therefore, the processing started from 5 kg of compost (remained from the last batch), and new waste is loaded daily, when the process enters the stand-by operational mode.

The experiment aimed at determining first the maximum daily variation of the mass of material entering the process (fresh waste), as well as the mass of waste found the equipment (waste being under processing).

The waste mass added daily in the process is weighed before filling the composter with a professional scale platform WB-700. The mass of material inside the equipment is evaluated daily (measuring the weight variation of the composting equipment).

The loading rate of organic material mass that was introduced into the composter is depicted in Figure 4.

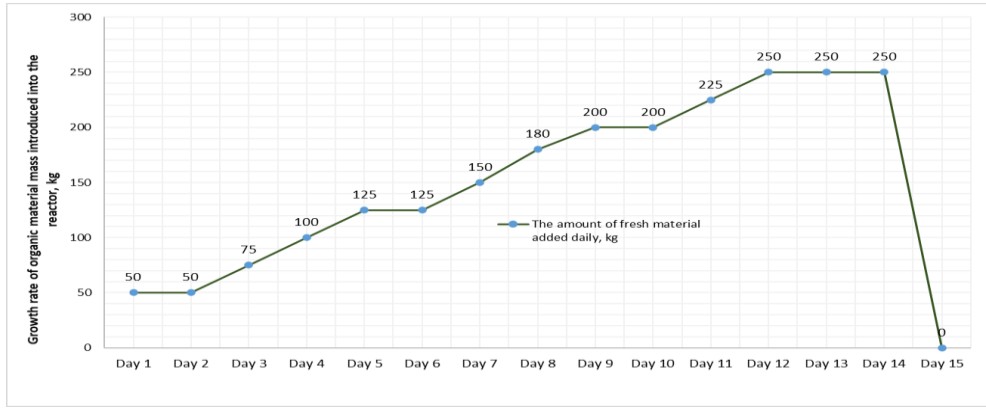

**Figure 4.** Proposed loading rate of the organic material mass that was introduced into the composting equipment.

The initial loading rate was determined empirically, considering the maximum amount of waste that can be added, in order to decrease the moisture below 20% within 24 h (until the next feeding).

### 2.4. Establishing the Operating Parameters for the Ceramic Heater, Air Pump, and the Mixing System

Experimental operating design is conducted with the help of a PLC, according to the data received from the sensors installed and associated with the established minimum and maximum limits.

During operation, the ceramic heater gradually raises the temperature of the organic material until the internal temperature of the mixture reaches 52 °C and then pauses, remaining off until the temperature drops below 48 °C. Although the operating range is set at 48–52 °C, due to thermal inertia the actual values may differ by 2–3°.

Mixing and aeration were set after a predefined operation strategy: 15 s mixing in one direction, then pause for 30 s, then start mixing in the opposite direction for 15 s and pause for 30 s. The blower (air pump) operates continuously for 2 min at maximum capacity, with one minute of pause, this cycle resuming independently of the operation of the other equipment under operation.

After 15 days, when the amount of material in the process has reached its maximum capacity and the dehydration process has been completed, the equipment automatically enters the "sanitization" state. In this state, the temperature of the ceramic heater rises to a constant value of 71 °C for one hour. The high treatment temperature for one hour aims to kill pathogens, plant seeds, and eliminate volatile compounds that could generate unpleasant odors. It should be noted that although thermophilic composting eliminated pathogens in the process itself, should be considered organic matter added on the last day, for which there is still the risk of containing contaminated meat products.

The 15 days frequency represents a strong asset for lowering waste collection transportation costs. Regarding the retail stores evaluated in Bucharest, the organic waste is collected twice a week; therefore, the use of the decentralized composting offers the possibility to eliminate up to 3 transports. In addition, a much smaller volume of waste will be sent to the sanitation company, leading to a lower waste management cost for the retail store. It is also important to note that at present, in Bucharest, the composting infrastructure of organic waste is much undersized. Therefore, at this moment, almost all organic waste is finally disposed to the city landfill, generating a high negative impact on the environment. The inoculum was not used in the study; however, a quantity of 5 kg of compost from the past batches remained in the equipment, taking over the role of a better multiplication of thermophilic bacteria.

The relevant indicators analyzed during composting were the total mass variation, moisture content, pH, pathogen presence, dry substance, loss by calcination, total organic carbon, total nitrogen, total phosphorus and potassium, calcium, magnesium, and potassium.

### 2.5. Sampling and Characterization Methods for Organic Wastes and Compost

The equipment functioning is significantly influenced by the organic waste that is being processed. Therefore, we aimed at determining the average quantities of organic wastes generated from retail, as well as the type and the characteristics of the wastes being produced.

In order to conduct research on the quantities and quality of waste being disposed, two types of food waste generators were considered: supermarkets and food court spaces.

Both locations have the highest amount of discarded food, due to their specificity. However, the supermarket generates wastes mainly due to food loss (expiration dates, altered vegetables/fruits, storage errors causing thawing, etc.), while food courts generate all variety of waste (wastes from food processing in the kitchen, leftovers food from the

customer menu, non-compliant or perishable foods, etc.). Considering the difficulties encountered by supermarkets in waste management, especially caused by the very high moisture level, our analysis focused on these retail stores.

A supermarket is a self-service shop offering a wide variety of food, which customers use most often as grocery shops, due to their smaller size.

Data on food waste quantities was processed based on four small size supermarket stores from Bucharest (sales area up to 20,000 square meters), the information being taken over between August and November 2021. Quantitative assessments were based on the reporting of waste generators, while qualitative assessments were performed based on direct measurements. In order to identify the main types of waste generated rates, the arithmetic mean was used to assess 10 samples, each sample having a volume of 60 L, for each of the 4 stores. The results section presents the arithmetic mean of the values recorded in the 4 retail stores.

The measured parameters of the equipment, using the sensors were recorded every 10 s, averaged for every 15 min period and saved in the data logger. Moisture content was measured by the oven-drying method (ASTM procedure D3173-73). A representative compost sample was placed in an air oven at 105 °C for 24 h, until a constant weight was achieved. The pH value was measured with the laboratory pH electrode, waste was mixed with de-ionized water in the proportion of 1:5 and it was stirred for one hour, pH being measured in the liquid phase. Moisture content was determined using the percent dry weight technique, which involved drying the probe at 105 °C to constant weight. During the maturation period, chemical parameters were determined by titrimetric method and flame photometry. Each measurement was performed for three times in order to reduce errors.

The input material generated by supermarket waste for composting will have a fairly similar content over time, with some variations influenced by the season.

### 2.6. Mechanism of Dehydration and Mineralization of Vegetable Waste That Contribute to Volume Decreasing

Conductive drying using ceramic heaters works by heating the water until it reaches the vapor stage. The mechanism of water removal by drying involves two simultaneous processes, namely, transfer of heat for the evaporation of water to the food and transport of the water vapors formed away from the food. Drying is, therefore, an operation based on simultaneous heat and mass transfer.

The moisture content of the samples was evaluated using an oven and standard hot air oven method (AOAC 2000), the initial moisture content being presented on dry basis in percentage.

The moisture ratio (MR) was calculated applying the Equation (1):

$$MR = \frac{M_d - M_e}{M_o - M_e}, \tag{1}$$

where *MR* is moisture ratio (dimensionless), *Md* is moisture content (gr water/gr dry solids), *Mo* is initial moisture content (gr water/gr dry solids), *Me* is equilibrium moisture content (gr water/gr dry solids).

When wastes are compressed, their volume is reduced, which is expressed in percentage and computed by calculating Volume Reduction *VR* (%), with the equation:

$$V_R = \frac{V_i - V_f}{V_f} \, 100, \tag{2}$$

where *Vi* is initial volume while *Vf* is the final volume.

The absolute minimum amount of energy that must be supplied for a drying process $Q_{v,min}$ is shown in Equation (3).

$$Q_{v,min} = W_v \, \Delta H_v, \tag{3}$$

where $\Delta H_v$ is the specific enthalpy of evaporation, J Kg$^{-1}$, $W_v$ mass flowrate Kg S$^{-1}$.

The compaction ratio (*CR*) of the waste is given in Equation (4):

$$C_R = \frac{V_i}{V_f},  \tag{4}$$

where $V_i$ = volume of waste before compaction, m$^3$ and $V_f$ = volume of waste after compaction, m$^3$.

### 3. Results

*3.1. Composition and Characterization of Food Waste Generated by Retail Stores (Category Small Supermarket)*

The four supermarkets were chosen as representatives for food retail stores in Bucharest, depending on several basic requirements. They had to fit into the category of small supermarket group, which are usually built-in residential areas (where the generation of unpleasant odors, leached, and pathogen is a major concern). For these stores, the space needed for waste storage is a problem, while the sales of fruits and vegetables are usually higher than average. Two of the supermarket stores were chosen in the city center, while the other two on the periphery area (to flatten the difference that may occur due to differences in living standards between residents). All stores maintain a strict waste reporting and have implemented selective waste collection in four fractions.

The waste from fresh fruits and vegetables was characterized by a low pH and a very high level of moisture, as expected. The analyzed samples showed that pH values varied between 4.25 and 5.12, while moisture was rated at 74–95%. The samples were taken randomly, the range of variation being caused by the proportions of each fruit or vegetable present in the waste.

Our concern was to analyze the waste disposed by the food waste generators that have implemented a high level of selective collection, to prevent contamination. In addition, thermophilic composting equipment works best for stores that discard more fruits and vegetables (having high humidity), considering that these wastes in particular causes biggest concern for retail stores.

Figure 5 describes the quantitative assessment results for the four supermarkets, according to the results reported annually by the management.

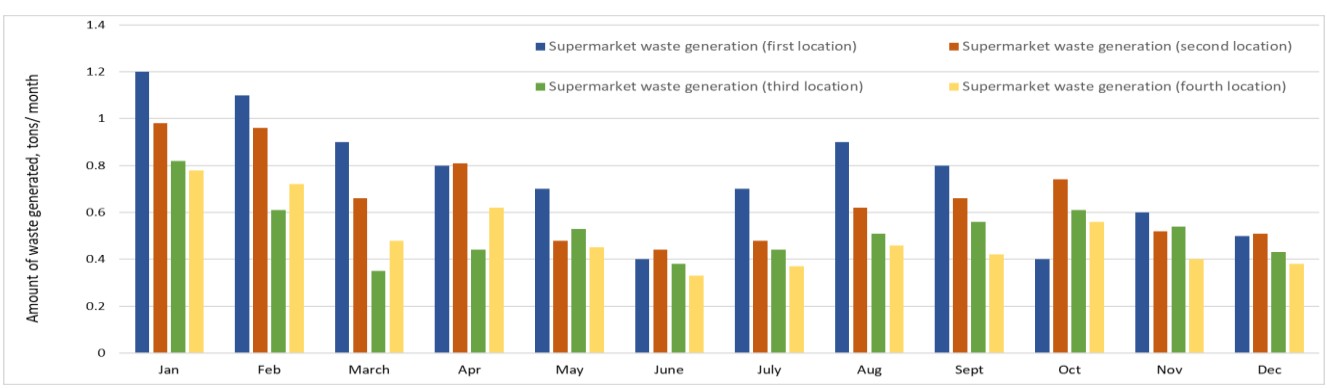

**Figure 5.** Quantitative assessment of food waste generated in four supermarkets from Bucharest, according to official reports submitted by management.

The quantities of food waste generated for year 2021 show a reduction compared to national statistics averages from the last three years by 20%. The COVID-19 pandemic has made customers to purchase larger food stocks, and on the other hand, the pressure of higher landfill costs has raised the waste-management efficiency. During March–July, the quantities of waste decrease as a result of smaller managed volumes, due to the competition with the local farmers that sell their own products in the agri-food markets.

However, in order to conduct an accurate evaluation for the composition of supermarket wastes, a qualitative analysis is also required. The qualitative analysis performed included identifying the percentage of fruits and vegetables from the total amount of

waste generated (Figure 6a) and the most common fruits and vegetables found in the mass of generated waste (Figure 6b).

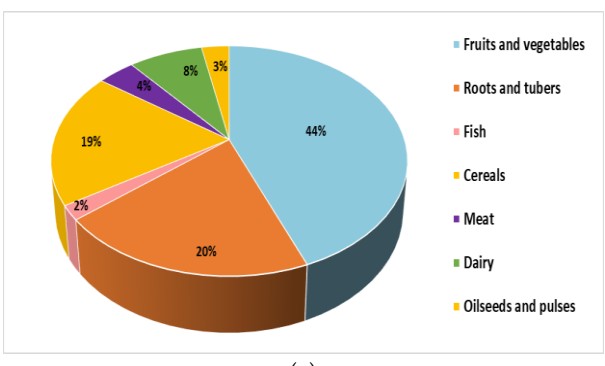
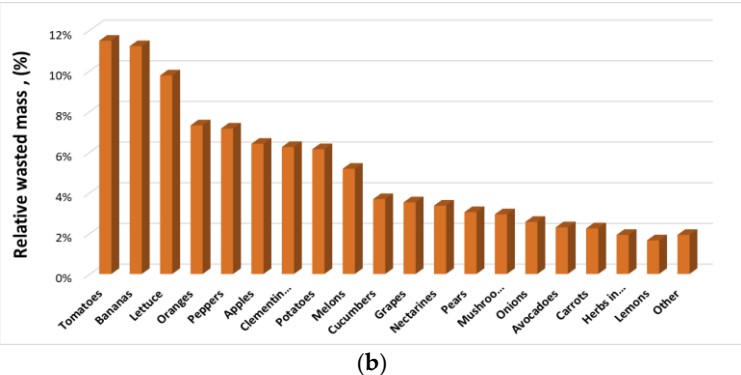

(**a**)           (**b**)

**Figure 6.** Assessment of the food waste generated in supermarkets: (**a**) classification of the main types of food waste; (**b**) identifying the most common fruits and vegetables found in the mass of generated waste.

Waste analysis showed that fruits and vegetables waste rated for over 40% of the total waste mass generated by the supermarkets. The analysis quantified the average values obtained for the four supermarkets. From Figure 6b can be deduced that first seven fruits and vegetables found in the supermarket wastes (tomatoes, bananas, salad, peppers, clementine, potatoes, and melons) represent over 70% of the total fruits and vegetables generated.

The average moisture content of fresh waste, before processing was 89%, a value validated by data obtained in other studies regarding fruits and vegetables humidity range [45].

### 3.2. Pre-Composting Process Performance

The processing aimed at decreasing final disposal costs (which were associated to the mass of waste reduction), eliminate pathogens, evaluate operational costs (correlated with energy consumed in the process), elimination of odors and leachate, and obtain a stable and inactive product with future potential to be used as fertilizer (considering a future stage of maturation).

### 3.2.1. Mass and Volume Reduction

The rapid decrease in mass and volume leads to the opportunity to process larger amounts of waste; therefore, this indicator is particularly important in decentralized composting. The decrease is caused by two phenomena, namely, the dehydration of vegetable waste and mineralization of organic matter.

The addition of new material in the process was based on the analysis of waste and its identified moisture. The moisture content of over 90% requires an effective water removal treatment, by applying high temperatures and air pump vapor evacuation, but without burning the material or inhibiting bacterial activity. It was found that daily feeding with fresh vegetable waste with high humidity no longer requires moistening processes. The process was not only aimed at waste dehydration, but also to create the needed conditions for the mineralization of organic matter and to favor the development of microorganisms that help the composting operation. The qualitative analysis of the waste and the optimal operation of the equipment were also considered.

The variation of the daily quantities of organic waste, remaining in the equipment during 15 days of processing in thermophilic regime with moisture extraction, associated to the proposed loading regime, is described in Figure 7.

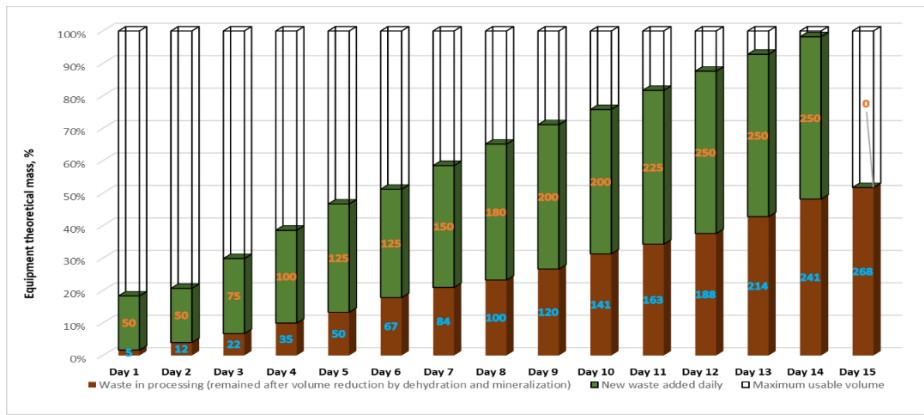

**Figure 7.** The variation of the daily quantities of organic waste, remaining in the equipment during 15 days of processing.

The brown color describes the mass of waste remaining in the equipment after each day of processing, while the green color represents the amount of fresh waste added daily. Equipment theoretical maximum mass capacity has been set to 500 kg (100%); however, the waste was gradually added so that mineralization and dehydration could take place in the best parameters.

The filling rate was set so that the maximum composter capacity to be reached in 14 days, while the treatment lasted another additional day without adding waste, in order to be able to treat the last input.

At the process initiation, a volume of 5 kg of compost have been added into the reactor, to provide the necessary microorganisms. In the first day, 50 kg of fresh waste material have been added to the process. In the second day, only 12 kg remained from the 55 kg processed, the rest being lost to evaporation, mineralization and air-pump evacuation. Every day the volume of waste added to the process increased (as can be seen in the graph), reaching 250 kg input in the last three days. The rate of weight loss has also increased over time, showing that the mineralization process takes place in good conditions and that reducing moisture by heating and dehydration does not burn the waste or kill microorganisms.

Using the data acquired above has been determined the amount of waste reduced daily as a result of processing, (as shown in Figure 8a), and then have been calculated the percentual daily distribution of the food waste, reporting the quantities added daily to the maximum mass of the equipment (Figure 8b). By linear regression, an equation is obtained, that can be used in practice by the operating personnel to easily customize the feeding technique for any other composter capacity that is using similar processing parameters.

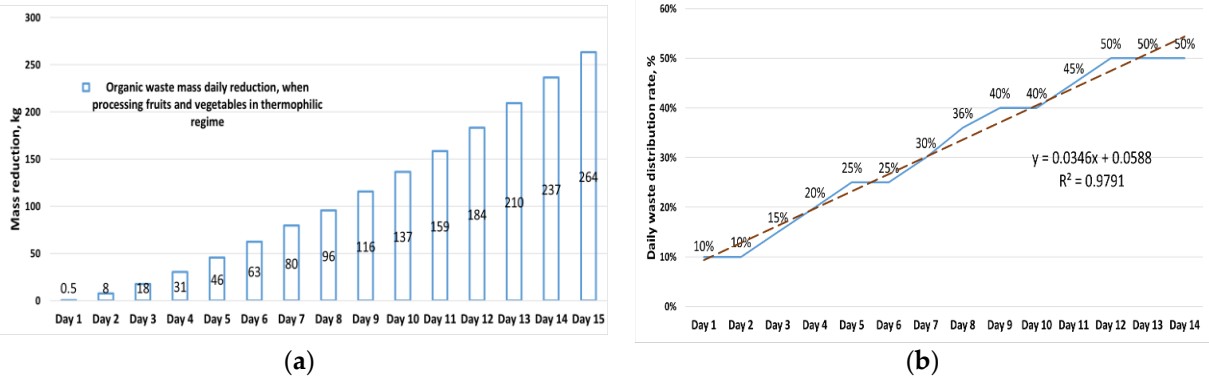

(**a**)　　　　　　　　　　　　　　　　　　　(**b**)

**Figure 8.** Calculation of the reduction rate of fruit and vegetable waste generated by retail stores: (**a**) determination of the organic waste mass daily reduction; (**b**) percentual daily distribution of food waste associated to maximum capacity of the equipment.

In the 15 days of processing, the total waste mass was reduced with over 1967 kg, from a total of 2235 kg processed waste; therefore, the cumulative reduction rate was exceeding 88%.

### 3.2.2. Temperature in the Process

Temperature plays an important role in composting, since the high temperature phase supports the growth of microbial activity, organic matter decomposes more easily and pathogenic organisms are more easily eliminated by exposure to heat. The testing phase was performed in real operating conditions, in an outdoor environment. During the considered processing time period, the average outside temperature was 27 °C during the day and 12 °C at night. The equipment was installed in an unheated hall, which dampens the effects of abrupt temperature changes. In order to maintain a high biodegradation rate, the temperature was electronically set to range between 48 °C and 52 °C.

In addition to the predetermined temperature control factors (ceramic heating system, that is automatically controlled by sensors, air blower operation, mixing and aeration operation, fresh material addition rate, and establishment of the sanitizing function), there are a number of other factors that influence the operation, which are not controlled. These factors are the external temperature, the humidity level of the fresh waste introduced, the volume and nature of the processed fruits and vegetables, and the temperature of the daily introduced waste.

Figure 9 depicts the temperature variation over the course of 15 days of composting process assessment, depicting the most efficient mode of operation, when fresh material is introduced exactly at the end of the previous day's treatment.

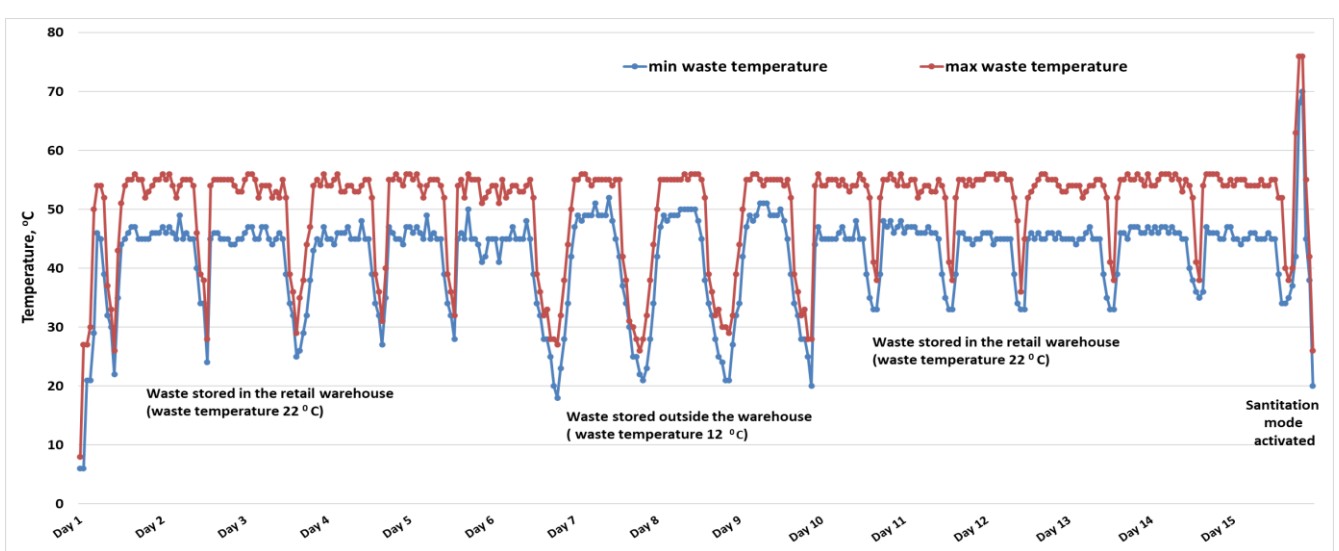

**Figure 9.** Temperature variation within 48 h in the composting equipment.

The temperature profile of the actively aerated reactor showed that the composting process varied in standard operating mode between 45–55 °C (a variation of 2–4° compared to the imposed parameters), due to the thermal inertia of the waste being processed.

The system is programmed to enter in a safe-mode, when moisture content of the waste drops below 20%, causing the heater to stop functioning, dropping the temperature to a minimum of 20 °C, in order to save energy and not excessively dehydrate the processed material. In addition, this action prevents some components of the composter from degrading, such as the ceramic heater. On the diagram, the process entering safe mode is described as a sharp drop in temperature, either until the waste meets the minimum temperature (20 °C) or until fresh material with higher humidity is supplied, and the process is therefore restarted.

It has been observed that when the waste is supplied directly from the retail store (days 1–7, 22 °C), the waste temperature does not influence in any way the operation of the equipment, the material reaching the processing temperature very quickly (maximum 15 min). On the other hand, when the waste is stored in bins outside the building during the night (days 8–10, 12 °C), arise a significant drop in the waste temperature.

On the 15th day of waste processing, the material sanitization phase was activated. Given that moisture content dropped below 20%, the moisture extractor stopped functioning, and the temperature of the waste rose to 75 °C for an hour. After pathogens removal, temperature slowly descend to 20 °C. The temperature drops slowly, because the material is relatively dry and the air pump is nor operating.

Figure 10 describes the characteristics of the material obtained after the 15 days showing the physical properties of the product obtained.

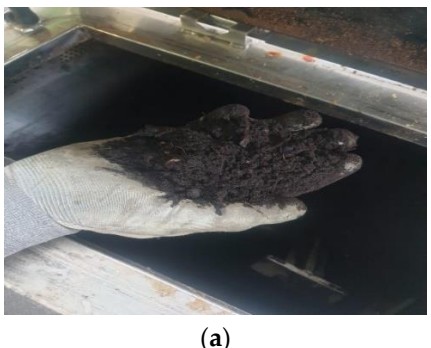
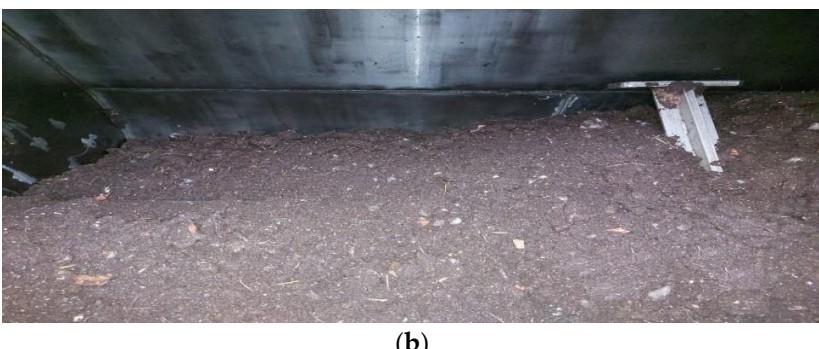

(**a**)　　　　　　　　　　　　　　　　　　　　　　　(**b**)

**Figure 10.** The characteristics of the compost obtained after the completion of the process; (**a**) the color and properties are specific to the soil; (**b**) some hard-coated waste will still remain visible in the material.

The material described in the Figure 10 showed a bulk density of 540 kg/m$^3$ and average granules diameter of 5.5 mm. We will not insist on the physical properties of the compost since the proposed technology gives a very good control of these parameters. By simply increasing the speed of the mixing blades, a smaller granulation can be obtained, while the bulk density can be easily modeled by a more intense water extraction.

### 3.2.3. Energy Consumption

The system adapts its mode of operation according to the temperature and moisture content of the waste, being significantly influenced by the outside temperature and the fruits and vegetables size. The heater maintains the temperature in the range of 45–55 °C, with more variations during sanitation phase, introduction of fresh waste and when entering power-saving mode. It is equipped with six ceramic heaters with a cumulated nominal power of 15 KW. The longest period of continuous operation of the heater takes place when loading the cold and high-humidity waste, on very cold days. For the rest of the operating period, the heater only adjusts the temperature of the material in the preset range. During the analyzed period (September–October) the average operating time of the heater was equivalent to 20 min of functioning at nominal power per hour.

The driving motor of the aeration, ventilation and shredding blades present a nominal power of 1.5 KW, and its average operating time is 20 min/h. The operation has been customized according to the waste introduced, which is very wet, and needs more dehydration and shredding. The air blower fan has a power of 0.37 KW, and the average operating time is 40 min/h. It is driven by the moisture detection sensors found in the waste, and the default operating program of 1 min functioning with 30 s pause.

The sensors, the automation and control system, and the alarming and protection mechanisms have a cumulative energy consumption of 0.1 KW, their operation being continuous. Therefore, the cumulative energy consumption of the equipment for the period

September–October with the processed material fruits and vegetables was 5.85 KWh. The total energy consumed by the equipment and dependencies in 14 days of operation was approximated to 2.75 MWh. A relevant example of the evolution of process characteristics during composting is shown in Table 1.

**Table 1.** Example of post-processing waste characteristics.

| | | | |
|---|---|---|---|
| Volume | Total fresh volume of waste introduced in processing | 2235 | Kg |
| | Output volume after processing | 268 | Kg |
| | Mass reduction | 1967 | Kg |
| | Reduction rate | 88 | % |
| Moisture | Initial waste moisture | 89 | % |
| | Output moisture (14 days) | 27 | % |
| | Output moisture (15 days) | 21 | % |
| | Moisture reduction rate | 90–92 | % |
| Pathogen presence (plant and animal) | | no pathogen or seeds present | |
| Dry substance | | 79 | % |
| pH | | 4.4 | unit |
| Loss by calcination (organic matter from dry substance) | | 78.05 | % dry substance |
| Total organic carbon | | 45.62 | % dry substance |
| Total nitrogen | | 2.75 | % dry substance |
| Total phosphorus | | 0.533 | % dry substance |
| Calcium | | 13,913 | mg/kg dry substance |
| Magnesium | | 1711 | mg/kg dry substance |
| Potassium | | 13,931 | mg/kg dry substance |

## 4. Discussion

The current research was initiated as a consequence of rapid increase of organic solid waste generation in Romania, which requires the development and testing of new operational and management technologies, especially oriented towards decentralized treatment. Thermophilic decentralized composting using automated composting equipment is a promising concept that has shown the potential to solve a wide range of solid waste management issues, in an economic and sustainable manner.

This study has a significant influence on urban waste management practices, since the expanding fluxes of organic waste are no longer properly managed worldwide, as a result of the continuous increase of food waste. Using an automated composting equipment and the thermophilic processing regime we addressed the most common limitations of traditional fruit and vegetable composting, such as poor compost quality, prolonged processing period, substantial nitrogen losses, GHG and different organic sulfur compound emissions, low pathogen inactivation effectiveness, and poor cleanliness.

### 4.1. Assessment of the Context of Romanian Composting Circumstances and the Opportunities Generated by the Proposed New Technology

Due to Romanian national legislations limitations and deficiencies regarding the municipal infrastructure for managing organic waste, the final destination of the compost discarded from municipal area is expected to be the landfill, at least until quality standards of the compost for various applications will be legislated. However, it should be noted that organic waste discarded from retail is a very good source for the production of high-quality compost, if the equipment operation is customized towards obtaining a high-quality product.

The study focused on the problems and needs encountered by retail stores; therefore, the first assessed assumption was targeted to the design of an operation strategy, that can reduce the cost of managing organic fruit and vegetable waste that generates problems due to leachate, unpleasant odors, potential for pathogens and large masses and volumes, which can be reduced without substantial effort on the part of employees.

### 4.2. Pre-Composting Process Performance Associated with Fruits and Vegetables Wastes
Waste Characteristics

Waste characterization carried out in the small sized supermarkets provides a qualitative and quantitative estimation for food waste in Bucharest retail sector, during the COVID 19 pandemic. During this period, the population changed behavior, buying more food than usually, and more than they needed, thus streamlining waste management in supermarkets. Although was recorded an increase in buying food products, reducing the level of waste disposal by 20% compared to the average of previous years. However, this cannot be considered a process efficiency, since some of the food waste has been transferred from the retail sector to household waste.

At retail level, fresh fruit and vegetables have been identified as the main contributor to the amount of the wasted material, rated at 45% of the total waste produced. Other research and statistics support the current findings. Literature data, shows that retail disposal rate from Italy stores was situated to 29% [46], and 34% [47], in United States 26% [48], in Austria 53% [49], and 46% in Sweden [50,51].

### 4.3. Moisture Content and Temperature in the Composting Process

The optimal moisture level for composting process must be an adjustment between providing sufficient moisture for microorganisms to develop and maintaining proper aerobic conditions and achieving dehydration that allows more material to be processed. The ideal moisture content in the initial material varies depending on the physical state and particle size, as well as the composting process utilized. The classic composting requires moisture content levels in the range of 40–65% of the compost weight, because microbial activity stops when the moisture content falls below 15% [51]. However, in the framework of continuous flow processing, with very moist material (case of fruits and vegetables), it is needed to eliminate moisture by dehydration up to lower values. Because of the large amounts of evaporation, the moisture level change continually throughout the composting period, slowing down the over-wetting phenomena that cause the process to delay, as well as anaerobic conditions appearance. The proposed technology gives the advantage of very precise customization of the resulting humidity level. However, the moisture should not be reduced below 15% during operation, as it could damage the material, even if a smaller volume could increase the operating theoretical efficiency.

Given the high importance of temperature in the process, it is preferable that the composting equipment to be positioned in a heated area, inside the buildings during the winter. The influence of the waste temperature was also found to be important, both for the time period needed for a complete cycle, as well as for the additional energy consumption. During the winter, the frozen waste introduced into the process showed at least an additional 45 min of heating to the maximum power. Storing wet waste outside the building in the summer can generate unpleasant odors and potentially hazardous leachate, so it is also to be avoided. In the experiments carried-out, no disturbances in the system operation were determined due to the temperature of the waste or to the environment, but only on the time and cost of processing. The proposed temperature variation regime proved to cover the daily necessary dehumidification levels, for the proposed feeding management (2235 kg in 14 days of treatment). The preset variation range was not always respected, especially due to the thermal inertia of the waste, which increased with the mass increasing. The sanitizing temperature (75 °C for an hour) has been shown to be

effective in destroying all pathogens and seeds present in the fresh material. One observation would be that the waste should be fed at supermarket temperature, given that outdoor storage can cause some reduction in treatment efficiency.

### 4.4. The Mixing and Homogenizing WASTE during Composting

When preparing the compost mix, the physical qualities of the materials must be always considered. Aeration, degradation potential, and the capacity to maintain aerobic conditions are affected by several physical factors, the three most important being the porosity, texture, and structure. In the case of fruits and vegetables, mixing is essential, because their specific composition has an impact on the resistance to airflow and the lack of air would quickly generate anaerobic processing. Failure to maintain adequate oxygen levels will result in odor generation and poor composting performance. The shape of the mixing blades not only mixes but also shreds the material, allowing air and microorganisms to enter the mixture more easily. The dehydration speed was also influenced by the mixing and the temperature level of the ceramic heaters.

### 4.5. Pathogen Removal

One of the advantages of the proposed waste management technique is the high degree of pathogen destruction, which is usually very difficult to control in traditional composting systems. Bacteria, viruses, fungi, and parasites are some of the pathogenic microorganisms that may be found in improperly treated compost. Despite the fact that parasites and viruses cannot replicate outside of their host, they may persist in the compost for long periods of time.

Heat, competition, nutritional depletion, antibiosis, and time are the factors that may all kill pathogens [52]. The majority of pathogens do not survive at the temperature variations defined inside the composting bioreactor. Under predetermined composting conditions, pathogens have been eliminated especially due to thermophilic regime.

Pathogens can also be eliminated as a result of competition for resources and space with the native microbial population. When we set the feeding rates, we considered this aspect, to maximize the chances of the development of microorganisms already present in the compost being used as inoculum. Therefore, the volume introduced aimed at increasing each day the volume of waste found in processing, for an efficient development of the composting bacteria already present in the compost. A higher amount of material would have increased the moisture content, giving the pathogens a chance to develop as well.

Pathogens are also killed by the rapid use of the accessible organic nutrients; therefore, the gradual addition of organic material may lead to the control of dangerous microorganisms. Well aerated and mixed material allow for better pathogen destruction compared to other techniques, because continuous mechanical turning allow the entire amount of material to be exposed to pathogen elimination conditions. In contrast, in the case of static piles, there will always be areas that have not reached the pathogen removal temperature (above or below the pile).

Unlike static traditional composting, composting in closed reactor also offers the advantage of reducing the risk of bioaerosol hazards. Bioaerosols are organisms or biological agents that are transported through the air and may cause health concerns, and in the case of the proposed technology this risk is minimized.

### 4.6. Energy Consumption and Expenses Efficiency

The proposed processing technique has a higher energy consumption than conventional composting systems. In order to make the process cost effective, the best objective would be to transform organic solid waste into valuable agricultural resources such as organic fertilizer or soil amendment, and to be able to achieve total pathogen inactivation and a high-volume reduction rate. For the Romanian case, due to legislative syncope, this cannot be done very easily at this moment. Therefore, the process must be carried out as

efficiently as possible, in the shortest possible time, in order to process as much waste as possible with minimum operating costs.

Composting options in closed processing bring many benefits in terms of ease of implementation in populated areas (due to lack of odors and, minimize the risk for the material to attracting insects), reduces the processing time of the organic material, and controls more efficiently the process of stabilization, sanitation, dehydration (with a positive impact on transport), and they may allow a superior quality of the products obtained.

The achieved results regarding the physical-chemical composition were adequate for the imposed conditions and for the expected results; however, a maturation stage would be necessary to increase the quality of the obtained product.

Considering the supermarkets objective to reduce expenses, a solution in this regard has been identified. By reducing the volume by 88% allowed the payment of smaller amounts to the sanitation company. However, the pilot considered only the fruits and vegetables wastes within this study (although the experiment may be extended to other types of organic waste), which would affect the cost-reduction percentage. Due to the fact that the sanitation company had to pick up the other organic waste, the transport cost was only affected by a 20% reduction. Another 20% reduction was the effect of project implementation on waste management, according to feedback received from supermarkets. The other positive effects such as the elimination of odors and leachate, the elimination of seeds and pathogens, the effect on customers by the adoption of innovative green technologies, could not be quantified without the extension of the pilot project for all types of organic waste discarded from supermarkets.

### 4.7. Final Considerations of the Proposed Technological Solution for Supermarkets

Since that the present study examined only the processing of fruits and vegetables in supermarkets (rather than all sorts of organic waste), using decentralized composting thermophilic regime, it is too early to make comparisons with other waste management techniques. In addition, supermarkets do not currently have the capacity to separate all organic waste on fractions, nor to eliminate all the contaminants, it is only the beginning of evaluating these techniques on a real scale.

Decentralized composting technology is still immature, and this results in a more difficult implementation in supermarkets chains. Composting in Romania is not subsidized or encouraged by the authorities. Although national targets set annually to reduce the amount of organic waste reaching the landfill, these targets are not achieved. A decentralized composting system present other additional environmental and social responsibility benefits that generates advantages to society and not only to the organic waste generator. It provides sustainable development advantages to the city, leading to progressive change toward low-carbon, resource-efficiency, resilience, and sustainable communities [52]. Studies confirm the reduction of the negative impact of inappropriate waste management landfill sites [53], reduce transportation and operational costs, and has significantly less CH4 gas emissions [54]. After a maturation process, the product being obtained using the technology presented in the article can be used as a soil amendment, bringing benefits on resource consumption and an increase in agricultural productivity [55,56]. Closed-loop circular food systems, which include composting and usage as a fertilizer, can result in 80% reduction of the energy required for vegetable production [57–59].

Associated with coherent legislative norms, to subsidize the reduction of greenhouse gases and to finance composting initiatives, decentralized approach would certainly increase globally, bringing undeniable benefits. Integrating wastes as agricultural fertilizers, may provide benefits not only for farmers, consumers, and waste management companies, but also would promote food sustainability and reduces food insecurity, especially in developing countries.

Depending on the objective pursued at the time of processing, if organic waste is mixed with other materials (plastic, glass, aluminum, batteries), may generate adverse

consequences on the quality or may even contaminate the compost. It is especially important to separate the waste into fractions before composting, even if some of the contaminating wastes may be degraded by bacteria within the thermophilic process.

In locations with increased humidity and cold weather, the outside temperature is a critical factor, especially if the composting equipment is located outdoors. It has been found that in winter, when the air humidity exceeds 90%, it is necessary to install an inlet dehumidification equipment, otherwise the very humid air entering the composter will condense on the cold walls of the equipment, increasing the liquid content inside. Excessive humidity can lead to anaerobic digestion, thus it's important to take preventative measures. If the installation of a dehumidifier is not possible, the waste can be processed using a mix of at least 50% sawdust or garden debris (dry). However, the addition of dry material to compensate for the high humidity is a temporary solution, the most appropriate measure is to position the equipment in a heated room with a low humidity.

Another circumstance when excessive humidity may affect the process is when large amounts of frozen wastes are being treated at once. The equipment works efficiently if the amount of frozen waste is added gradually, so as not to produce a very large amount of liquid inside. Programming a strict waste addition schedule, as proposed in this present article, is not always feasible in practice. If the waste is added chaotically in the reactor, and in addition the nitrogen/carbon ratio is not taken into account (which would regulate the process somehow), then the phenomenon of hardening of the organic processed material may occur. In extreme situations, the phenomenon may lead to damaging the equipment due to the excessive force that must be exerted in the process.

## 5. Conclusions

Decentralized thermophilic composting has proven to be an effective management option for wet wastes, in particular for fruits and vegetables, reducing processing time, decreasing mass, volume, and moisture content, while having a superior control over the process. The new composting technology showed an efficient control of processing parameters, such as temperature, moisture content of the material, mixing level, and airflow rate, leading to a better control of the process, a better treatment of waste, and total reduction of producing unpleasant odors or pathogenic agents.

Since the process uses high amounts of energy, several quality indicators must be followed in order to ensure that the compost has a high quality and may be utilized as fertilizer. Future research will be focused on establishing the best operating mode for obtaining high-quality compost and to add other types of food waste to the process.

**Author Contributions:** Conceptualization, F.N., I.S., H.V., A.G., T.A., O.L.T. and D.M.C.; methodology, F.N., C.S. and V.N.V.; software, F.N., O.L.T. and T.A.; validation, O.L.T., D.M.C. and V.N.V.; formal analysis, I.S., H.V., A.G. and V.N.V.; investigation, F.N., D.M.C., T.A. and C.S.; resources, T.A., O.L.T. and V.N.V.; data curation, V.N.V.; writing—original draft preparation, F.N., I.S. and H.V.; writing—review and editing, F.N., O.L.T., T.A., D.M.C., C.S., I.S. H.V. and V.N.V.; visualization, O.L.T. and V.N.V.; supervision, F.N., O.L.T. and T.A.; project administration, F.N., O.L.T. and T.A.; funding acquisition, F.N., O.L.T. and T.A. All authors have read and agreed to the published version of the manuscript.

**Funding:** This paper was supported by a grant offered by the Romanian Minister of Research as Intermediate Body for the Competitiveness Operational Program 2014-2020, call POC/78/1/2/, project number SMIS2014 + 136213, acronym METROFOOD-RO.

**Institutional Review Board Statement:** Not applicable.

**Informed Consent Statement:** Not applicable.

**Data Availability Statement:** Not applicable.

**Conflicts of Interest:** The authors declare no conflict of interest.

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
