# Peer review of "Decentralized Processing Performance of Fruit and Vegetable Waste Discarded from Retail, Using an Automated Thermophilic Composting Technology"

_sustainability, doi:10.3390/su14052835_

Round 1
Reviewer 1 Report
Abstract is very generalized. Please provide some important facts and figures of your findings in this section.
Introduction section is very long and requires considerable reduction in details provided.
Line211-223: the information is more relevant to the study objectives at the end of introduction as it should be not the part of materials and methodology section.
Details regarding methods/ tests used for data analysis are missing.
Line 768-69: Need better structuring of the sentence.
As mentioned by the authors about the cost effectiveness of the method and indicated the economical nature of the process but they did not provide a comparison with other available methods. how much cost effective the process is?
Please provide facts and figures to support your stance.
The authors may consider in reducing the length of all the section of the manuscript as it seems quite long at the moment.
Author Response
Dear evaluator,
Thank you for taking your time to make such an extensive assessment. Please find below the changes made to the document, according to your comments.
1. Abstract is very generalized. Please provide some important facts and figures of your findings in this section.
We have changed the abstract considering the main findings and reducing the degree of generalization.
2. Introduction section is very long and requires considerable reduction in details provided.
The size of the article has been reduced, especially the introductory section, however other reviewers considered necessary to add new information, which affected the structure and size of the paper.
We also numbered the Introductory section so that it could be better analyzed, we are sorry we couldn't reduce the text more.
3. Line211-223: the information is more relevant to the study objectives at the end of introduction as it should be not the part of materials and methodology section.
As recommended, we have deleted the text between lines 211-223.
4. Details regarding methods/ tests used for data analysis are missing.
We clarified in detail the methods / tests used for data analysis
5. Line 768-69: Need better structuring of the sentence.
The mentioned sentence was replaced with a new paragraph, and the benefits were justified with bibliographic citations.
6. As mentioned by the authors about the cost effectiveness of the method and indicated the economical nature of the process but they did not provide a comparison with other available methods. how much cost effective the process is? Please provide facts and figures to support your stance.
The cost - efficiency part is quite complex and cannot be included in this article, especially in the context of energy costs rising at European level, due to the conflict in Ukraine. However, we have introduced some indicative information on efficiency and cost reduction between lines 793-794.
7. The authors may consider in reducing the length of all the section of the manuscript as it seems quite long at the moment.
We tried to reduce the size as much as possible. We deleted some paragraphs, instead we had to introduce some additional ones during the evaluation.
Thank you!
Reviewer 2 Report
The manuscript describes the decentralized processing performance of fruit and vegetable waste discarded from retail, using an automated thermophilic composting technology. The manuscript is interesting and could be published after major assessments:
- Abstract – Too general. Provide ‘real values’ in the main findings. Provide clear objectives in the abstract. Re-write your abstract.
- Introduction too long. Shorten in not more than 5 paragraphs. Include/highlight only relevant literature review.
- Introduction – objectives stated not clear. Suggested to provide aim of study. Specific objective by numbering i.e. (1) To…… (2) To…..etc
- Section 2.1 – your explanation seems to explain about research objectives. Supposely in introduction section? Not understood very well. Or you may provide flow chart of overall methodology, so that reader could understand.
- Line 252-270. The label should be in the caption (Figure 2), instead you write in paragraph.
- In section 2.2, any references citation when you design the equipment? What are the main reasons/considerations for each part you designed for composting equipment?
- Figure 4 – How you decide proposed loading rate? Any calculation involves? Any reference based?
- Line 369 – Explain why you decide not to use inoculum?
- Section 2.3 - Economic evaluation of full-scale implementation: How do you calculate the costs of implementing decentralized composting for retail? What do you meant by current situation – explain clearly? Any equation involves in the calculation. Provide clear equation and how to calculate.
- What are the criteria when you select four supermarkets from Bucharest…explain in methodology.
- Line 481-482. Could you explain the mechanism dehydration of vegetable waste and mineralization of organic matter that contribute to decrease in volume? Explain in terms of factors that control the composting process.
- Provide equation in your method. i.e. how you calculate equipment theoretical mass, etc.
- Check font type for Table 1.
- Major concern for discussion section – only 8 references citation. How do you compare your results/discussion with other findings from other researchers? More citation is needed to validate and scientifically discussed your results. Seems like the author explained using his/her opinion without referring to other researches.
- How about conclusion section to verify the objectives stated in the introduction? Missing? Provide one paragraph of conclusion. Short conclusion but sweet.
- Check your languages/grammar. Suggest proofreading.
Author Response
Dear evaluator,
Thank you for taking your time to make such an extensive assessment. Please find below the changes made to the document, according to your comments.
- Abstract – Too general. Provide ‘real values’ in the main findings. Provide clear objectives in the abstract. Re-write your abstract.
We have changed the abstract considering the main findings and reducing the degree of generalization.
- Introduction too long. Shorten in not more than 5 paragraphs. Include/highlight only relevant literature review. Introduction – objectives stated not clear.
The size of the article has been reduced, especially the introductory section, however other reviewers considered necessary to add new information, which affected the structure and size of the paper.
We also numbered the Introductory section so that it could be better analyzed, and reformulated the objectives. We are sorry we couldn't reduce the text more.
- Section 2.1 – your explanation seems to explain about research objectives. Supposely in introduction section? Not understood very well. Or you may provide flow chart of overall methodology, so that reader could understand.
- Many paragraphs from section 2.1 have been reformulated, in order to make it clearer to the reader. This section is part of the methodology, and explains why no other composting technique has been approached, in contrast to the classic composting methods.
- Please find between lines 204-205 the flow chart of overall methodology
- Line 252-270. The label should be in the caption (Figure 2), instead you write in paragraph.
- As suggested, we introduced the labels in the caption. The paragraph, containing the descriptive elements was moved below the figure.
- In section 2.2, any references citation when you design the equipment? What are the main reasons/considerations for each part you designed for composting equipment?
The prototype was built by improving a model that is marketed by a partner company of our institute. Because the prototype has superior features compared to the model sold in the market, the manufacturing company preferred not to provide constructive details of the prototype.
- Figure 4 – How you decide proposed loading rate? Any calculation involves? Any reference based?
The initial loading rate was determined empirically, considering the maximum amount of waste that can be added, in order for the humidity to decrease below 20% with-in 24 hours (until the next feeding). This paragraph was introduced in the paper.
- Line 369 – Explain why you decide not to use inoculum?
We have introduced an explanation why the inoculum was not used in this study, between lines 178-182: “Therefore, when designing the methodology and the automatic decentralized composting system used in the processing of fruit vegetable waste, we consider especially the needs of the stores and less the compost quality. The retailers refused to use inoculum and enzymes, since the compost superior quality does not bring any benefits to the stores, and increase their acquisition and operating costs.”
- Section 2.3 - Economic evaluation of full-scale implementation: How do you calculate the costs of implementing decentralized composting for retail? What do you meant by current situation – explain clearly? Any equation involves in the calculation. Provide clear equation and how to calculate.
We have removed section 2.3. At the moment, the conflict in Ukraine has had a major effect on rising energy prices. In the last 4 months, the price of energy and methane gas has increased 3-4 times. Therefore, at this time the costs would no longer be fair for this technology.
- What are the criteria when you select four supermarkets from Bucharest, explain in methodology.
We have introduced the reasons why the 4 supermarkets were chosen as relevant for the food retail stores in Bucharest: “The 4 supermarkets were chosen as representatives for food retail stores in Bucharest, depending on several basic requirements. They had to fit into the category of small supermarket, which are usually built in residential areas (where the generation of unpleasant odors, leached and pathogen is a major concern). For these stores, the space needed for waste storage is a problem, while the sales of fruits and vegetables are usually higher than average. Two of the supermarket stores were chosen in the city center, while the other two on the periphery area (to flatten the difference that may occur due to differences in living standards between residents). All stores maintain strict waste reporting and have implemented selective waste collection in four fractions.”
- Line 481-482. Could you explain the mechanism dehydration of vegetable waste and mineralization of organic matter that contribute to decrease in volume? Explain in terms of factors that control the composting process. Provide equation in your method.
A series of equations have been introduced that describe the dehydration part. (Section 2.6, Lines 398 – 452)
- Check font type for Table 1.
We modified the font in the table, according to the specifications of the journal.
- Major concern for discussion section – only 8 references citation. How do you compare your results/discussion with other findings from other researchers? More citation is needed to validate and scientifically discussed your results. Seems like the author explained using his/her opinion without referring to other researches.
More references have been introduced.
- How about conclusion section to verify the objectives stated in the introduction? Missing? Provide one paragraph of conclusion. Short conclusion but sweet.
We removed the conclusions in the previous edition so as not to risk repeating the material, but on your advice, we reintroduced them short. Thank you.
Check your languages/grammar. Suggest proofreading.
The manuscript has been revised in terms of linguistics and some typing errors have been corrected.
Thank you!
Reviewer 3 Report
The submitted work evaluates the performance of a decentralized composting approach using an automated thermophilic composter. The study, in general, is very interesting. However, several issues need to be addressed prior to its further processing. Please check the specific comments below:
- The language of the manuscript requires significant improvement.
- The title is very long. Consider shortening the title.
Abstract:
- In the abstract, the authors put much emphasis on the study background rather than the results. The abstract needs to be rewritten highlighting the findings and practical implications of the research.
- L20: Remove the first ‘-’ from the word composting.
Introduction:
- L44: Cite some examples.
- L48-49: Not clear. Consider rewriting the statement.
- L50-51: Mention some of the difficulties.
- L53: Consider replacing the word ‘In addition’ with ‘Furthermore’.
- In general, the introduction contains way too much information which is sometimes redundant. The authors should focus on presenting the key information and research gaps.
Materials and methods:
- This study is based on the automated operation. However, not that much information is available regarding the automated process in the materials and methods section.
- Figure 2: For a better understanding of the readers, the authors should consider incorporating the name of the different parts in the figure.
- Section 2.2.1: Consider rephrasing the section heading. At present, it seems like a statement.
- Check the section numbers.
- Section headings need to be rewritten as a title.
Results:
- The overall presentation of the results is fine. The authors mentioned they performed an economic evaluation of full-scale implementation in the materials and methods section, which is missing in the results.
- Add section numbers.
Discussion:
- The results are well discussed; however, it would be better to include issues that the authors faced during the study. Indication of such issues will help researchers to conduct further studies.
Author Response
Dear evaluator,
Thank you for taking your time to make such an extensive assessment. Please find below the changes made to the document, according to your comments.
The submitted work evaluates the performance of a decentralized composting approach using an automated thermophilic composter. The study, in general, is very interesting. However, several issues need to be addressed prior to its further processing. Please check the specific comments below:
- The language of the manuscript requires significant improvement.
The manuscript has been revised in terms of linguistics and some typing errors have been corrected.
- The title is very long. Consider shortening the title.
We reduced the title size as suggested.
Abstract:
3. In the abstract, the authors put much emphasis on the study background rather than the results. The abstract needs to be rewritten highlighting the findings and practical implications of the research.
4. L20: Remove the first ‘-’ from the word composting.
Thank you for the observation, we edited the abstract to be more results-oriented and less general. We also eliminated the error in the word "composting".
Introduction:
5. L44: Cite some examples.
Added examples as suggested. L43-45
6. L48-49: Not clear. Consider rewriting the statement.
We have modified the sentence to make it clearer.
7. L50-51: Mention some of the difficulties.
A paragraph has been introduced, that lists some of the most common difficulties. L54-57
8. L53: Consider replacing the word ‘In addition’ with ‘Furthermore’.
The word has been replaced.
9. In general, the introduction contains way too much information which is sometimes redundant. The authors should focus on presenting the key information and research gaps.
The size of the article has been reduced, especially the introductory section, however other reviewers considered necessary to add new information, which affected the structure and size of the paper.
Materials and methods:
10. This study is based on the automated operation. However, not that much information is available regarding the automated process in the materials and methods section.
Although we did not focus on automation characteristics, we have introduced some elements that describe the benefits of system automation. L 289-295
11. Figure 2: For a better understanding of the readers, the authors should consider incorporating the name of the different parts in the figure.
As suggested, we introduced the labels in the caption. The paragraph, containing the descriptive elements was moved below the figure.
12. Section 2.2.1: Consider rephrasing the section heading. At present, it seems like a statement.
The section heading have been rephrased.
13. Check the section numbers.
The changes have been made.
14. Section headings need to be rewritten as a title.
Paragraph and section numbering has been rewritten, as you suggested
Results:
15. The overall presentation of the results is fine. The authors mentioned they performed an economic evaluation of full-scale implementation in the materials and methods section, which is missing in the results.
The cost - efficiency part is quite complex and cannot be included in this article, especially in the context of energy costs rising at European level, due to the conflict in Ukraine. However, we have introduced some indicative information on efficiency and cost reduction between lines 793-794.
We have removed section 2.3. that mentioned the economic aspects, detailed information on costs will be deepened in future research.
16. Add section numbers.
Section numbers have been added to the subsections, as suggested.
Discussion:
17. The results are well discussed; however, it would be better to include issues that the authors faced during the study. Indication of such issues will help researchers to conduct further studies.
These aspects have been added in section 4.7., starting with the lines L 840.
Thank you very much!
Round 2
Reviewer 2 Report
Check this - Line 157 - kinf? kind?
Reviewer 3 Report
After the revision, the manuscript as a whole is now more scientifically sound and logically described.